# Reframing the National Institute on Minority Health and Health Disparities Research Framework: Strengthening the Behavioral Domain with the Inclusion of Psychological Factors

**DOI:** 10.3390/ijerph22070992

**Published:** 2025-06-24

**Authors:** Caleb Esteban, Normarie Torres-Blasco, Alíxida Ramos-Pibernus

**Affiliations:** 1Queer Biopsychosocial Health Laboratory (The Queer Lab), School of Behavioral and Brain Sciences, Ponce Health Sciences University, Ponce 00716, Puerto Rico; 2Ponce Research Institute, Ponce 00716, Puerto Rico; normarietorres@psm.edu (N.T.-B.); aliramos@psm.edu (A.R.-P.); 3Hispanic/Latinx Intervention Development for Psychosocial Empowerment Lab (HIPE Lab), School of Behavioral and Brain Sciences, Ponce Health Sciences University, Ponce 00716, Puerto Rico; 4Health Equity Research Lab (HER Lab), School of Behavioral and Brain Sciences, Ponce Health Sciences University, Ponce 00716, Puerto Rico

**Keywords:** NIMHD Research Framework, behavioral domain, psychological domain, psychological factors, psychological determinants of health

## Abstract

The National Institute on Minority Health and Health Disparities Research Framework (NIMHD-RF) provides a multidimensional structure to examine health disparities across domains and levels of influence. While influential, its current Behavioral Domain centers on observable behaviors and underrepresents key psychological factors and determinants that shape health outcomes among minoritized populations. This gap limits the framework’s capacity to account for complex factors such as internalized stigma, identity-related stress, and cultural processes that significantly contribute to mental health disparities. In this viewpoint, we propose an adaptation of the Behavioral Domain into a Psychological/Behavioral Domain to better reflect the interconnected psychological, biological, sociocultural, and environmental factors influencing health. The revised domain incorporates psychological vulnerabilities, coping strategies, and identity-based stressors across all levels of influence, from individual to societal, and acknowledges macro-level processes such as structural stigma and inequitable policies. This reframing emphasizes that behaviors are shaped by psychological experiences and systemic inequities, not merely individual choice. By explicitly integrating psychological factors and determinants, the framework becomes more robust in guiding culturally responsive, equity-driven research and interventions. This adaptation aims to enhance the framework’s utility in mental health disparities research and to support efforts to achieve health equity for historically underserved populations.

## 1. NIMHD Research Framework and Limitations of the Behavioral Domain

The National Institute on Minority Health and Health Disparities Research Framework (NIMHD-RF) is a tool to better understand and address health disparities through a multidimensional lens [1]. This multidimensional model is a merger of the National Institute of Aging Health Disparities Research Framework and the Socioecological Model, resulting in a matrix with two axes: domains of influence and levels of influence on health. The domains are listed from micro to macro including biological, behavioral, physical/built environment, sociocultural environment, and healthcare system. At the same time, the levels are listed micro to macro including individual, interpersonal, community, and societal. However, while the NIMHD Research Framework is a groundbreaking tool for understanding health disparities, its current Behavioral Domain inadequately addresses the psychological factors and determinants that underlie health inequities. This omission limits the framework’s ability to capture complex influences such as stigma, identity, and cultural factors, which are critical to understanding and addressing health outcomes for minoritized populations.

The NIMHD-RF serves as a vital framework for investigating and addressing health disparities among minoritized populations [1]. However, its current Behavioral Domain focuses primarily on observable behaviors (e.g., smoking, substance use) while overlooking the broad psychological factors and determinants of health outcomes. In this reflection, psychological determinants are conceptualized as psychological conditions or processes that may contribute directly to health disparities (e.g., anxiety, depression, stress), while psychological factors are broader in scope and may include variables that influence, moderate, or result from these health disparities, making them more context-dependent and potentially bidirectional in their relation to health outcomes. For example, research shows that internalized stigma among sexual and gender minority (SGM) individuals is strongly associated with increased rates of depression and anxiety, which in turn contribute to health-risk behaviors like substance use and poor medication adherence [2]. Behavior, while visible, reflects only the surface of a complex interplay between psychological, sociocultural, and environmental factors. This oversight limits the framework’s ability to fully capture the underlying mechanisms driving health disparities. Health disparities are rarely the result of a single factor; rather, they emerge at the intersection of multiple systems of oppression, including racism, heterosexism, classism, and coloniality, that compound and interact with individual-level psychological processes. To fully address minority health, research frameworks must account for the dynamic interplay between structural and psychological factors and determinants. To address this critical gap, we propose reframing the Behavioral Domain into a Psychological/Behavioral Domain. This extension is essential for fostering a more comprehensive understanding of the psychological factors and determinants (e.g., internalized stigma) that significantly shape health outcomes among minority and minoritized populations [3,4].

## 2. Distinguishing Behavioral and Psychological Determinants

The American Psychological Association defines ‘behavioral’ as any observable action or measurable function in response to specific stimuli [5]. In contrast, psychology encompasses the study of the mind and behavior, exploring complex processes such as biological, cognitive, emotional, personal, and social factors underlying human actions. Similarly, the field of psychiatry, as defined by the American Psychiatric Association [6], goes beyond behavior by focusing on the diagnosis, treatment, and prevention of mental, emotional, and behavioral disorders. While behavioral research has been foundational to psychological investigation, psychological factors and determinants encompass a far more expansive and intricate landscape. Psychology examines the interactions between biological, psychological, sociocultural, and environmental factors. Recognizing the breadth and multifaceted nature of psychology is essential to advancing a holistic understanding of minority health and addressing disparities.

In 2022, Alvidrez and Barkdale [7] presented an adaptation of the NIMHD-RF for multidimensional mental health disparities. Their framework incorporated both general social determinants of health and those specific to mental health, for a comprehensive approach that simultaneously examines various domains and levels of influence. However, they highlighted that addressing behaviors alone fails to capture the complexity of mental health disparities, underscoring the need for a deeper exploration of psychological factors such as identity, stigma, discrimination, and cultural influences. Despite this, their adaptation continues to use ‘behavioral’ as the domain name, even though its recommendations extend far beyond the behavioral concept.

Furthermore, definitions of mental health emphasize a broader foundation that includes emotions, thinking, communication, learning, resilience, and relationships, factors that extend beyond observable behaviors [3,5]. While behaviors are one component, they cannot fully encapsulate the psychological factors and determinants that shape mental health. A behavioral reductionism perspective risks oversimplifying these complexities, which are critical to understanding and addressing health disparities effectively.

The field of psychology, along with other mental health disciplines, provides critical insights into the complex interplay of biological, psychological, sociocultural, and environmental factors that shape health outcomes. To advance health equity, researchers must adopt innovative approaches that address critical issues, for example, the stigma associated with psychological diagnoses, particularly among ethnic minorities such as Hispanic/Latinx populations. By recognizing the broader spectrum of influences beyond observable behaviors, the NIMHD framework can more effectively capture the complexity of mental health disparities. These disparities are often driven by psychological factors and determinants, including identity, mental health disorders, stigma, perceived discrimination, and cultural factors [8].

## 3. Proposed Psychological/Behavioral Domain

Our proposed adaptation of the Behavioral Domain to a Psychological/Behavioral Domain addresses critical gaps in the NIMHD Research Framework by incorporating a broader range of psychological factors and determinants and contextual influences (see Table 1).

At the individual level, it includes intersectionality factors such as psychological vulnerabilities (e.g., anxiety, depression, stress), identity (e.g., racial/ethnic, sexual orientation, gender identity), health behaviors (e.g., smoking, substance use), and coping strategies (e.g., mindfulness, emotional suppression), which influence health outcomes beyond observable actions. The interpersonal level highlights the role of relationships, including the impact of discrimination, relationship functioning, family dynamics, and school or workplace functioning, where experiences such as bullying or unsupportive family interactions could exacerbate health disparities. At the community level, factors like stigma and community functioning (e.g., access to mental health services and social resources) capture the influence of collective environments, particularly in resource-limited or biased settings. The societal level underscores macro-level influences, such as structural stigma (e.g., institutional discrimination against non-monogamous relationships), mental health laws and policies (e.g., disparities in insurance coverage), and environmental justice policies (e.g., pollution disproportionately affecting minority communities), which create inequities in mental health outcomes.

This adaptation recognizes that behavior alone cannot account for the complex interplay of biological, psychological, sociocultural, and environmental factors shaping health disparities. By broadening the scope to explicitly include psychological factors and determinants, the framework provides a more comprehensive understanding of mental health disparities. This nuanced approach enables researchers and policymakers to develop culturally responsive and equity-driven interventions that better address the needs of minoritized populations, particularly in mental health research. Expanding the Behavioral Domain to reflect this complexity is essential for fostering health equity and addressing the root causes of disparities.

## 4. Conclusions

Culturally sensitive frameworks are of key importance, as they highlight differences in experience and impact in the context of minority groups. This holistic approach allows for a more nuanced understanding of health disparities and facilitates the development of targeted interventions that address the diverse needs of minoritized communities. Additionally, by recognizing the role of psychology, and other mental health-related disciplines, in shaping health-related behaviors, attitudes, and beliefs, the framework can pave the way for more effective strategies to promote mental health, well-being, and quality of life among marginalized populations.

This reconceptualization also holds important implications for public policy, as it supports the design of evidence-informed policies that prioritize psychological well-being, address stigma-related barriers, and promote culturally grounded mental health interventions. By incorporating these psychological dimensions into public health strategies, policymakers can more effectively target the root causes of disparities and implement inclusive approaches that advance structural change.

This adaptation does not aim to invalidate or disrupt existing research employing the NIMHD-RF. In fact, many researchers, particularly in mental health, have already been incorporating psychological factors and determinants under the label of the Behavioral Domain. Our proposal brings conceptual clarity by explicitly naming and distinguishing psychological factors and determinants, emphasizing the importance of accurate language and visibility. Rather than conflicting with past work, this reframing provides a more precise structure for future studies and supports the evolution of a framework that better reflects the realities of mental health and health disparities. It offers a transition that strengthens, rather than displaces, existing efforts in health disparities research.

In the pursuit of health equity, the NIMHD-RF should include an improved domain that recognizes how psychological factors and determinants impact the health of minoritized groups at all levels of influence. Moreover, it gives the same weight to psychological vulnerabilities as the current framework recognizes for biological vulnerabilities. Embracing the complexity of psychology aspects and recognizing the limitations of a purely behavioral lens has the potential to transform the field of minority health by providing researchers with more comprehensive tools to expand the understanding of the etiology of the health disparities while at the same time improving the overall health of communities at large.

## Figures and Tables

**Table 1 ijerph-22-00992-t001:** Adaptation of the Behavioral Domain of the NIMHD Research Framework with integrated psychological factors and determinants.

Domains of Influence	Levels of Influence			
	Individual	Interpersonal	Community	Societal
**Psychological/** **Behavioral**	*Psychological* *Vulnerabilities* *Identity* Health Behaviors Coping Strategies	*Discrimination* *Relationship* *Functioning* Family Functioning School/Work Functioning	*Stigma* Community Functioning	*Structural Stigma* *Mental Health Laws/Policies* *Environmental Justice Policies*

Notes: *Italics* = new and adapted suggested factors. This table is an adaptation of the behavioral domain from Table 1 in Alvidrez and Barksdale [7], which itself is an adaptation of the NIMHD Research Framework. It focuses exclusively on the behavioral domain, acknowledging that the full framework includes additional domains (e.g., biological, sociocultural, environmental, and healthcare system) that are not represented here. What distinguishes this adaptation is that it integrates both psychological and behavioral domains of influence across the four levels of influence, offering a more comprehensive lens for understanding and addressing mental health and health disparities.

## Data Availability

No new data were created or analyzed in this study.

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
