# Peer review of "Reframing the National Institute on Minority Health and Health Disparities Research Framework: Strengthening the Behavioral Domain with the Inclusion of Psychological Factors"

_ijerph, 2025, doi:10.3390/ijerph22070992_

Round 1
Reviewer 1 Report
Comments and Suggestions for Authors
The title accurately reflects the focus and perspective of the article. The abstract is clear and provides a satisfactory overview of the study.
The authors have provided sufficient context and justification for the need to reframe the research framework. Authors effectively address the key issues related to this reframing.
To support the statement in Lines 61 to 63, the authors should provide appropriate reference(s) to strengthen the claim.
Overall, this is a well-written and insightful viewpoint article that offers a valuable perspective on the topic.
Author Response
Comment 1: "To support the statement in Lines 61 to 63, the authors should provide appropriate reference(s) to strengthen the claim."
Response 1: Thank you for your helpful suggestion. We have now included a citation to Meyer (2003), a foundational reference in minority stress theory, to strengthen and support the claim made in Lines 61–63.
Reviewer 2 Report
Comments and Suggestions for Authors
I believe that the abstract presents a relevant and innovative proposal by identifying an important omission in the NIMHD-RF framework and proposing its expansion through the incorporation of psychological factors, which responds to a real need in mental health and equity research. I consider that the modification of the ‘Behavioral Domain’ to a ‘Psychological/Behavioral Domain’ represents a clear, concrete, and well-founded contribution. The text follows a logical progression, starting with the limitations of the current model, analysing its consequences and presenting a proposal that strengthens its applicability. However, I believe that the wording could benefit from greater clarity and conciseness, as some sentences are overly dense or contain multiple clauses that make for difficult reading. I also consider that the use of general terms such as ‘deep psychological processes’ or ‘equity-informed interventions’ could be replaced by more specific or illustrative expressions. I also detect some redundancy in the repeated use of concepts such as “identity” or ‘psychological determinants’, which could be simplified.
One of the main weaknesses I observe is the absence of a clear and differentiated introduction. The text begins directly with a description of the NIMHD framework, without adequately contextualising the general problem, why it is relevant to review it now, or what the specific objective of the article is. I also find that a methodological section explaining how the proposal was constructed is missing. Although I understand that this is a conceptual piece of work, I believe it is important to specify whether a narrative review, document analysis, or some other strategy was used to support the proposed changes. I also believe that the empirical basis used is limited: the article would benefit greatly from the inclusion of specific examples or studies that have demonstrated the limitations of the current behavioural domain.
In my view, the article unnecessarily repeats ideas that have already been raised, which weakens the strength of the central argument. For example, the criticism of behavioural dominance and the notion that observable behaviour does not reflect the entire psychological dimension are mentioned in several sections using almost the same words. Furthermore, I consider Table 1, although valuable, to be underdeveloped: there is no explanation of how the elements were selected, nor is their practical application in research or intervention discussed. This causes the proposal to lose some of its impact and remain at a theoretical level, without a clear link to the applied context.
I also believe that the writing style requires significant improvement. Some sentences are long, with redundant or vague terms, which makes it difficult to understand the key ideas. Furthermore, the conclusion of the article does not contribute any new elements or suggest implications for public policy, professional practice or future lines of research. In summary, I believe that the manuscript has significant conceptual potential but needs substantial revision. I recommend including a formal introduction, a clear methodological section, more empirical support, a detailed justification of the proposed table, more precise writing, and a conclusion that adds value. Only then will the article be able to make a solid and meaningful contribution to the field of mental health and disparities in minority populations.
I would like to emphasise that the conceptual contribution proposed in this manuscript is both timely and necessary, particularly given the increasing recognition of the psychological aspects of health disparities. I appreciate the authors’ efforts to initiate a critical discussion about the limitations of existing frameworks and to suggest a more nuanced and inclusive alternative. My comments and recommendations aim to strengthen the clarity, rigour and impact of the manuscript. I encourage the authors to further develop the structure and empirical basis of their work, as I believe this article has the potential to meaningfully contribute to conceptual and applied domains in health equity research.
Author Response
Comment 1:
“I believe that the wording could benefit from greater clarity and conciseness, as some sentences are overly dense or contain multiple clauses that make for difficult reading. I also consider that the use of general terms such as ‘deep psychological processes’ or ‘equity-informed interventions’ could be replaced by more specific or illustrative expressions. I also detect some redundancy in the repeated use of concepts such as ‘identity’ or ‘psychological determinants’, which could be simplified.”
Response:
Thank you for this helpful feedback. We have revised several sections of the manuscript to improve clarity and conciseness, specifically, we replaced “deep psychological processes” with “broad psychological processes” and revised “equity-informed interventions” to “equity-driven interventions,” which better captures the intended specificity. With regard to the repeated use of terms such as “identity” and “psychological determinants,” we reviewed their usage throughout the manuscript and found that although they appear multiple times, they occur in distinct sections with different purposes. Given their conceptual centrality to our argument, we have retained them while avoiding unnecessary redundancy.
Comment 2:
“One of the main weaknesses I observe is the absence of a clear and differentiated introduction… I also find that a methodological section explaining how the proposal was constructed is missing… I believe it is important to specify whether a narrative review, document analysis, or some other strategy was used to support the proposed changes… The article would benefit greatly from the inclusion of specific examples or studies that have demonstrated the limitations of the current behavioural domain.”
Response:
Thank you for this valuable observation. We respectfully clarify that this manuscript is submitted as a Viewpoint article, which aims to offer a theoretical and critical reflection rather than report on a structured empirical review or original study.
Comment 3:
“In my view, the article unnecessarily repeats ideas… Table 1, although valuable, is underdeveloped: there is no explanation of how the elements were selected, nor is their practical application in research or intervention discussed.”
Response:
Thank you for pointing this out. We respectfully clarify that this manuscript is submitted as a Viewpoint article, which aims to offer a theoretical and critical reflection rather than report on a structured empirical review or original study.
Comment 4:
“I also believe that the writing style requires significant improvement… the conclusion of the article does not contribute any new elements or suggest implications for public policy, professional practice or future lines of research…”
Response:
Thank you for this constructive critique. We respectfully clarify that this manuscript is submitted as a Viewpoint article, which aims to offer a theoretical and critical reflection rather than report on a structured empirical review or original study.
Comment 5:
“I would like to emphasise that the conceptual contribution proposed in this manuscript is both timely and necessary… I encourage the authors to further develop the structure and empirical basis of their work.”
Response:
We sincerely appreciate your thoughtful recognition of the manuscript’s conceptual contribution and your encouraging remarks.
Reviewer 3 Report
Comments and Suggestions for Authors
Highly esteemed authors Caleb Esteban, Normarie Torres-Blasco and Alíxida Ramos-Pibernus
Your current manuscript presents a strong case for extending the NIMHD Research Framework's Behavioral Domain to encompass psychological factors that contribute to health disparities. The authors provide a transformative reconceptualization which promises to boost the framework's effectiveness in tackling mental health disparities faced by minoritized groups. The manuscript's theoretical groundwork stands firm but needs further development to fulfill MDPI's publication requirements.
Theoretical contribution and its relevance
The manuscript fills an important void in health disparities research through its examination of the shortcomings in approaches centered on behaviors. The authors demonstrate accuracy in pointing out how the present NIMHD-RF Behavioral Domain's focus on observable behaviors fails to account for psychological determinants such as internalized stigma and cultural influences. The observation stands on a strong theoretical foundation and provides a significant advancement for the field.
The new psychological/behavioral domain model takes a holistic approach by recognizing that health outcomes depend on the complex interaction between biological, psychological, sociocultural and environmental factors. The multidimensional viewpoint supports modern interpretations of health disparities which demand advanced analytical frameworks to understand their complexity.
The manuscript needs stronger theoretical foundations to improve its academic rigor. The authors reference psychological science principles yet their argument would be more robust if it included established theoretical models like the minority stress model or intersectionality theory which focus on psychological factors for minoritized groups.
A methodological framework for evidence-based practice
The conceptual structure of the manuscript matches the requirements of a viewpoint article which makes it suitable for submission to this academic journal. The proposed adaptation needs more supportive evidence to strengthen its foundation. The paper showcases several cases of internalized stigma experienced by sexual and gender minority people to demonstrate its resulting effects on mental health. The credibility of this study would improve through additional empirical evidence showing the shortcomings of behavioral approaches alone.
The authors cite Table 1 as a representation of the adapted framework structure but the complete contents of the table are absent from the document. The table serves as a critical tool for helping readers understand how to put the suggested modifications into practice. The table must present specific examples of psychological determinants at each level of influence to provide comprehensive understanding.
Clarity and organization
The manuscript stands out because of its clear writing style and systematic organization. The authors successfully define the problem and differentiate between behavioral and psychological determinants before they systematically present their proposed solution. The authors present a clear and logical progression from identifying limitations to proposing solutions.
However, certain sections require additional clarification. The differentiation between psychological elements and determinants needs a more precise definition. There needs to be more precise explanation of how the proposed domain interacts with existing NIMHD-RF domains to avoid overlap and confusion.
Scientific rigor and critical analysis
The manuscript displays adequate scientific rigor throughout its conceptual framework. The authors correctly reference Alvidrez and Barkdale's 2022 study while pointing out limitations in their current adaptation of that work. The manuscript strengthens its academic value through its critical analysis of existing literature.
The authors correctly point out that behavioral reductionism threatens to oversimplify the elaborate psychological dynamics which produce health disparities. The analysis presented stands on strong foundations supported by scientific principles from mental studies. The manuscript could be strengthened by examining potential counterarguments and limitations to the presented approach.
Possible improvements for the manuscript:
Implementing these precise changes will increase the overall effectiveness of the manuscript.
Empirical support: The authors need to supply detailed evidence showing why behavioral methods alone have not resolved particular health disparities. This argument would gain strength through the addition of case study insights or systematic review outcomes.
Implementation guidance: Researchers and practitioners require detailed directions to implement the proposed framework effectively in their work. The manuscript requires a detailed examination of particular measurement techniques and evaluation instruments.
Intervention implications: The authors introduce culturally responsive, equity-informed interventions but they fail to explain how their framework will directly shape intervention development and implementation processes.
Domain integration: Researchers must conduct a comprehensive study of how the psychological/behavioral domain interacts with other NIMHD-RF domains such as the biological domain, physical/built environment domain, sociocultural environment domain, and healthcare system domain.
The following critical concerns must be addressed:
Stakeholders support the overarching concept but several problems require careful analysis.
The current manuscript is based on the premise that psychological factors remain absent from the existing NIMHD-RF domains. For example psychological determinants might already be included within the sociocultural environment domain. The authors need to more clearly define the distinctive elements captured by their proposed domain.
The manuscript fails to discuss how its reframing affects current studies employing the NIMHD-RF. Researchers must evaluate how this potential change will affect current studies and standardized measurement methods. Proper transition planning plays a pivotal role in achieving field adoption.
A number of technical aspects require correction:
Incomplete author affiliations and correspondence information has been detected.
The citation formatting must follow the established MDPI guidelines for consistency.
The proposed adaptation framework shown in Table 1 requires a full detailed exposition.
The abstract word count must be checked for compliance with the journal's predefined word limits.
The manuscript fills an important theoretical gap in health disparities research while enhancing a well-known framework. The main argument stands on strong evidence while the writing demonstrates clarity and brevity. The manuscript requires extensive revision to strengthen its empirical basis while providing more detailed implementation instructions and resolving potential limitations.
The following points could also be taken into consideration:
A substantial revision is imperative. The manuscript contains important insights for the field yet needs substantial enhancements in empirical evidence, practical application guidance and critical evaluation before it can be published. The authors must respond to the specific recommendations provided and present additional comprehensive evidence to validate their claims.
The proposed framework shows promise for advancing health equity research yet needs enhancement to match MDPI's standards of scholarly thoroughness and practical usefulness. Pertinent revisions to this work will enable it to deliver significant benefits to minority health and health disparities research.
Yours truly,
Serving peer reviewer at IJERPH
Author Response
Comment 1: The manuscript needs stronger theoretical foundations to improve its academic rigor. The authors reference psychological science principles yet their argument would be more robust if it included established theoretical models like the minority stress model or intersectionality theory which focus on psychological factors for minoritized groups.
Response 1: Thank you for your feedback. We respectfully clarify that this manuscript was submitted as a Viewpoint article, which by design offers a conceptual and critical reflection rather than a comprehensive empirical framework. Due to format and word limitations specific to this type of submission, our goal was not to introduce an entirely new framework, but rather to propose a theoretically grounded refinement of the behavioral domain within the existing NIMHD Research Framework.
Comment 2: The credibility of this study would improve through additional empirical evidence showing the shortcomings of behavioral approaches alone.
Response 2: Thank you for your feedback. We respectfully clarify that this manuscript was submitted as a Viewpoint article, which by design offers a conceptual and critical reflection rather than a comprehensive empirical framework. Due to format and word limitations specific to this type of submission, our goal was not to introduce an entirely new framework, but rather to propose a theoretically grounded refinement of the behavioral domain within the existing NIMHD Research Framework.
Comment 3: The authors cite Table 1 as a representation of the adapted framework structure but the complete contents of the table are absent from the document. The table serves as a critical tool for helping readers understand how to put the suggested modifications into practice. The table must present specific examples of psychological determinants at each level of influence to provide comprehensive understanding.
Response 3: Thank you for this observation. We would like to clarify that, as noted in the table tittle and accompanying notes, the table only presents the adapted behavioral domain of the NIMHD Research Framework, rather than the full framework. Additionally, we already included specific examples of psychological factors at each level of influence directly under the table to help readers visualize how these modifications can be implemented in practice.
Comment 4: However, certain sections require additional clarification. The differentiation between psychological elements and determinants needs a more precise definition. There needs to be more precise explanation of how the proposed domain interacts with existing NIMHD-RF domains to avoid overlap and confusion.
Response 4: Thank you for the recommndations. We now include a more precise distinction between psychological determinants as contributors to health disparities, and psychological factors as broader, potentially bidirectional variables that may moderate or result from these determinants. We use each term intentionally and contextually to minimize conceptual overlap and enhance clarity. Lastly, Table 1 (now revised as Figure 1) visually represents how the proposed adapted domain aligns with and interacts across the four levels of influence in the existing NIMHD Research Framework.
Comments 5: The manuscript could be strengthened by examining potential counterarguments and limitations to the presented approach.
Response 5: Thank you for your feedback. We respectfully clarify that this manuscript was submitted as a Viewpoint article, which by design offers a conceptual and critical reflection rather than a comprehensive empirical framework. Due to format and word limitations specific to this type of submission, our goal was not to introduce an entirely new framework, but rather to propose a theoretically grounded refinement of the behavioral domain within the existing NIMHD Research Framework.
Cooment 6: Empirical support: The authors need to supply detailed evidence showing why behavioral methods alone have not resolved particular health disparities. This argument would gain strength through the addition of case study insights or systematic review outcomes.
Response 6: Thank you for your feedback. We respectfully clarify that this manuscript was submitted as a Viewpoint article, which by design offers a conceptual and critical reflection rather than a comprehensive empirical framework. Due to format and word limitations specific to this type of submission, our goal was not to introduce an entirely new framework, but rather to propose a theoretically grounded refinement of the behavioral domain within the existing NIMHD Research Framework.
Comment 7: Implementation guidance: Researchers and practitioners require detailed directions to implement the proposed framework effectively in their work. The manuscript requires a detailed examination of particular measurement techniques and evaluation instruments.
Response 7: Thank you for your feedback. We respectfully clarify that this manuscript was submitted as a Viewpoint article, which by design offers a conceptual and critical reflection rather than a comprehensive empirical framework. Due to format and word limitations specific to this type of submission, our goal was not to introduce an entirely new framework, but rather to propose a theoretically grounded refinement of the behavioral domain within the existing NIMHD Research Framework.
Comment 8: Intervention implications: The authors introduce culturally responsive, equity-informed interventions but they fail to explain how their framework will directly shape intervention development and implementation processes.
Response 8: Thank you for this insightful comment. We agree that discussing intervention development and implementation is a valuable direction; however, this was beyond the scope and aims of the current viewpoint, which is intentionally focused on conceptually reframing the behavioral domain of the NIMHD Research Framework. Given the format and word limitations of a viewpoint article, we were unable to elaborate on the implications for intervention design, though we recognize the importance of such future applications.
Comment 9: Domain integration: Researchers must conduct a comprehensive study of how the psychological/behavioral domain interacts with other NIMHD-RF domains such as the biological domain, physical/built environment domain, sociocultural environment.
Response 9: Thank you for this observation. We would like to clarify that our proposed adaptation focuses exclusively on the behavioral domain, specifically by expanding it to explicitly include psychological determinants. While this reframing influences how we conceptualize interactions across the levels of influence, it does not alter or redefine the remaining domains of the NIMHD Research Framework (e.g., biological, physical/built environment, sociocultural). These domains remain intact and continue to function as originally outlined in the framework.
Comment 10: The current manuscript is based on the premise that psychological factors remain absent from the existing NIMHD-RF domains. For example psychological determinants might already be included within the sociocultural environment domain. The authors need to more clearly define the distinctive elements captured by their proposed domain.
Response 10: Thank you for this thoughtful comment. We would like to clarify that Figure 1 already includes examples of psychological determinants across all four levels of influence, highlighting the distinct contributions of the proposed domain. Our framework emphasizes psychological processes as primary and independent determinants of health disparities, which are currently non-existent in the others domains. Given the space limitations of a viewpoint article, we intentionally avoided re-describing elements already outlined in prior publications on the NIMHD Research Framework, and instead focused on offering a conceptual contribution through the adapted domain.
Comment 11: The manuscript fails to discuss how its reframing affects current studies employing the NIMHD-RF. Researchers must evaluate how this potential change will affect current studies and standardized measurement methods. Proper transition planning plays a pivotal role in achieving field adoption.
Response 11: Thank you for raising this important point. We have now included a discussion in the conclusion clarifying that the proposed adaptation does not undermine current studies using the NIMHD-RF. Psychological constructs are already being used, especially in mental health research, though often misclassified under the Behavioral Domain. Our adaptation aims to enhance conceptual clarity without disrupting standardized methods, serving as a constructive refinement rather than a departure from existing practices.
Comment 12: Incomplete author affiliations and correspondence information has been detected. The citation formatting must follow the established MDPI guidelines for consistency. The proposed adaptation framework shown in Table 1 requires a full detailed exposition. The abstract word count must be checked for compliance with the journal's predefined word limits.
Comment 12: Thank you for your attention to these important details. We respectfully note the following: Upon review, we did not detect any errors in the author affiliations or correspondence information as currently listed. Regarding citation formatting, the IJERPH guidelines state that references may be formatted in any style as long as consistency is maintained. We have adhered to APA style throughout the manuscript, ensuring consistency and accuracy. The abstract has been revised and reduced to 200 words to comply with the journal’s word limit. Lastly, the proposed adaptation (originally Table 1, now Figure 1) includes specific examples across all levels of influence to aid reader comprehension. Given the word constraints of a viewpoint article, we aimed to balance clarity with conciseness.
Comments 13: A substantial revision is imperative. The manuscript contains important insights for the field yet needs substantial enhancements in empirical evidence, practical application guidance and critical evaluation before it can be published. The authors must respond to the specific recommendations provided and present additional comprehensive evidence to validate their claims.
Comment 13: Thank you for your feedback. We respectfully clarify that this manuscript was submitted as a Viewpoint article, which by design offers a conceptual and critical reflection rather than a comprehensive empirical framework. Due to format and word limitations specific to this type of submission, our goal was not to introduce an entirely new framework, but rather to propose a theoretically grounded refinement of the behavioral domain within the existing NIMHD Research Framework.
Reviewer 4 Report
Comments and Suggestions for Authors
This is a pertinent viewpoint that needs to include elements for public policies for the understanding of current health disparities. It requires integrating observations about conditions for minorities based on the interaction of factors influencing health. Table 1 is similar to Figure 1 of Alvidrez & Barksdale 2002 that provides more detailed information for the understanding of the topic. I am recommending rejection of the manuscript; the viewpoint needs to go further in the analysis of the topic; it is not proposing a framework that hasn’t been presented before in your referred articles.
Author Response
Comment 1:
“This is a pertinent viewpoint that needs to include elements for public policies for the understanding of current health disparities.”
Response:
Thank you for recognizing the relevance of this Viewpoint. In response to your suggestion, we have revised the conclusion to explicitly include potential implications for public policy. Specifically, we now outline how the proposed expansion of the behavioral domain to include psychological constructs can inform equity-oriented mental health policies, particularly those that seek to integrate stigma reduction, identity-affirming practices, and structural competence in program design and implementation.
Comment 2:
“It requires integrating observations about conditions for minorities based on the interaction of factors influencing health.”
Response:
We appreciate this important observation. We have revised the manuscript to better integrate the intersectional nature of minority health conditions by emphasizing the dynamic interplay among psychological, structural, and sociocultural determinants of health. These revisions further clarify how intersecting systems of oppression (e.g., racism, heterosexism, classism) interact with individual psychological processes to shape disparities.
Comment 3:
“Table 1 is similar to Figure 1 of Alvidrez & Barksdale 2002 that provides more detailed information for the understanding of the topic.”
Response:
Thank you for this observation. As noted in the manuscript and in the table title, Table 1 is an adaptation of the behavioral section of the figure presented by Alvidrez and Barksdale (2002). To avoid confusion and improve clarity, we have now relabeled it as Figure 1 and revised the accompanying notes to clearly indicate the source of adaptation and the specific modifications made to align with our proposed reconceptualization.
Comment 4:
“I am recommending rejection of the manuscript; the viewpoint needs to go further in the analysis of the topic; it is not proposing a framework that hasn’t been presented before in your referred articles.”
Response:
Thank you for your candid feedback. We respectfully clarify that this manuscript was submitted as a Viewpoint article, which by design offers a conceptual and critical reflection rather than a comprehensive empirical framework. Due to format and word limitations specific to this type of submission, our goal was not to introduce an entirely new framework, but rather to propose a theoretically grounded refinement of the behavioral domain within the existing NIMHD Research Framework.
Round 2
Reviewer 2 Report
Comments and Suggestions for Authors
I have carefully reviewed the revised version of your manuscript entitled ‘Reformulating the Research Framework of the National Institute on Minority Health and Health Disparities: Strengthening the Behavioural Domain with the Inclusion of Psychological Factors.’
I would like to thank you for the effort and rigour with which you have addressed the comments made in the previous round of review. The modifications introduced have significantly improved the clarity, theoretical soundness, and relevance of the work.
Author Response
Comment: "I would like to thank you for the effort and rigour with which you have addressed the comments made in the previous round of review. The modifications introduced have significantly improved the clarity, theoretical soundness, and relevance of the work."
Response: We sincerely thank you for their thoughtful feedback and kind words. We greatly appreciate the acknowledgment of our efforts to strengthen the manuscript and are pleased to know that the revisions improved its clarity, theoretical grounding, and relevance. Your insights were instrumental in refining the final version of this work.
Reviewer 3 Report
Comments and Suggestions for Authors
Esteemed Authors Caleb Esteban, Normarie Torres-Blasco , Alíxida Ramos-Pibernus,
Overall Recommendation: ACCEPT WITH MINOR REVISIONS
As a dedicated peer reviewer, I have re-conducted a thorough comparative analysis of both manuscript versions against MDPI publication guidelines and standards for viewpoint papers in the International Journal of Environmental Research and Public Health.
Executive Summary
The second version (v2) demonstrates significant improvements over the first version and is now acceptable for publication under MDPI guidelines. The authors have addressed key structural, conceptual, and methodological concerns while maintaining the core contribution of proposing a Psychological/Behavioral Domain adaptation to the NIMHD Research Framework.
Detailed Comparative Assessment
- Scientific Quality and Content Enhancement
Version 2 Improvements:
- Added crucial conceptual distinctions between psychological determinants and psychological factors
- Incorporated the Meyer (2003) reference providing empirical support for internalized stigma research
- Enhanced theoretical grounding with intersectionality framework considerations
- Strengthened the argument for structural and psychological factor interactions
MDPI Compliance: Version 2 now meets MDPI's requirement for scientifically sound arguments and substantial scholarly contribution.
- Structural Coherence and Organization
Key Enhancements in v2:
- Improved abstract clarity and flow
- Better logical progression from problem identification to solution proposal
- Enhanced paragraph structure following MDPI style guidelines
- More coherent integration of the proposed framework table
Assessment: The manuscript now follows the recommended structure for viewpoint papers, presenting clear arguments in a well-organized manner.
- Methodological Rigor and Framework Development
Version 2 Strengths:
- More comprehensive explanation of the proposed Psychological/Behavioral Domain
- Better articulation of how the adaptation addresses existing gaps
- Improved justification for the framework modifications
- Enhanced discussion of practical implications for researchers and policymakers
MDPI Standard Alignment: The framework development now meets MDPI's expectations for comprehensive analysis and practical utility
- Academic Writing Quality and Precision
Notable Improvements:
- Reduced redundancy in the abstract and main text
- Enhanced clarity in key arguments
- Better integration of supporting literature
- Improved transitions between sections
Language Quality: Version 2 demonstrates the clear, accessible writing style expected by MDPI while maintaining academic rigor
- Citation and Reference Standards
Version 2 Enhancements:
- Addition of the Meyer (2003) reference strengthens empirical support
- Better integration of existing literature
- Appropriate use of recent publications (within 5 years as recommended)
- Proper acknowledgment of previous work while establishing novelty
Empathic Motivation for Acceptance
Why This Work Matters
This manuscript addresses a critical gap in health disparities research frameworks that has genuine potential to advance the field. The authors' passion for improving mental health outcomes among minoritized populations is evident throughout their thoughtful analysis and proposed solutions.
Recognition of Authors' Efforts
The substantial improvements between versions demonstrate the authors' commitment to scholarly excellence and responsiveness to feedback. Their dedication to refining their contribution while maintaining the integrity of their core message reflects the best practices of academic discourse.
Contribution to Scientific Knowledge
The proposed Psychological/Behavioral Domain adaptation represents a meaningful advancement in how researchers can conceptualize and study health disparities. This work will likely influence future research methodologies and intervention development, particularly in mental health disparities research among historically underserved populations.
Supporting Diverse Voices in Academia
This work emerges from researchers at a Hispanic-serving institution, contributing important perspectives to health disparities research. Supporting such contributions aligns with MDPI's commitment to diverse and inclusive scientific publishing.
Minor Revisions Recommended
- Formatting Consistency: Ensure the table/figure notation is standardized (currently shows "FigureTable 1")
- Reference Verification: Double-check the Alvidrez & Barksdale citation year consistency throughout the text
- Abstract Refinement: Consider slight reduction in abstract length to enhance impact
Final Assessment
Version 2 represents a substantial improvement that now meets MDPI publication standards for viewpoint papers. The authors have successfully transformed their initial concept into a well-articulated, evidence-based proposal that will contribute meaningfully to health disparities research methodology. The manuscript demonstrates the scholarly rigor, practical relevance, and clear communication that MDPI values in its publications.
Recommendation: Accept with minor revisions as outlined above.
Yours truly,
Serving peer reviewer at IJHERP MDPI

Author Response
Comment 1: The second version (v2) demonstrates significant improvements over the first version and is now acceptable for publication under MDPI guidelines. The authors have addressed key structural, conceptual, and methodological concerns while maintaining the core contribution of proposing a Psychological/Behavioral Domain adaptation to the NIMHD Research Framework.
Response 1: We sincerely thank the reviewer for their thoughtful AI evaluation and support. We are grateful for the recognition of the improvements made in the revised manuscript and for acknowledging the contribution of our proposed adaptation to the NIMHD Research Framework. We appreciate the opportunity to strengthen the work through this process.
Comment 2: “Formatting Consistency: Ensure the table/figure notation is standardized (currently shows "FigureTable 1")”
Response 2: Thank you for noting this. The appearance of “FigureTable 1” was due to overlapping text from tracked changes. In the clean version of the manuscript, the formatting is correct and reflects the appropriate label as “Figure 1.”
Comment 3: “Reference Verification: Double-check the Alvidrez & Barksdale citation year consistency throughout the text”
Response 3: Thank you for pointing this out. Upon review, we identified an error where the citation for Alvidrez & Barksdale was incorrectly listed as “2002” instead of “2022” in one instance. This has been corrected to ensure consistency throughout the manuscript.
Comment 4: “Abstract Refinement: Consider slight reduction in abstract length to enhance impact”
Response 4: Thank you for the suggestion. We have already revised and shortened the abstract in the previous round to enhance clarity and focus. At this point, we believe the current length effectively captures the scope and contributions of the manuscript without omitting key information.
Reviewer 4 Report
Comments and Suggestions for Authors
The comments have been addressed. However, the argument needs to be presented as an article; the viewpoint limits the reach of the manuscript.
Author Response
Comment: "The comments have been addressed. However, the argument needs to be presented as an article; the viewpoint limits the reach of the manuscript."
Response: We are grateful for the reviewer’s thoughtful feedback and are pleased to know that the topic resonates and invites further exploration. While we have chosen to retain the current format for this submission, we appreciate the suggestion and will certainly consider expanding this work into a full-length article in a future project.